# Hybrid Metal-Organic Frameworks/Carbon Fibers Reinforcements for Additively Manufactured Composites

**DOI:** 10.3390/nano13050944

**Published:** 2023-03-05

**Authors:** Marwan Al-Haik, Suma Ayyagari, Yixin Ren, Andrew Abbott, Bing Qian Zheng, Hilmar Koerner

**Affiliations:** 1Department of Mechanical Engineering, Kennesaw State University, Marietta, GA 30060, USA; 2Department of Aerospace Engineering, Embry-Riddle Aeronautical University, Daytona Beach, FL 32114, USA; 3Materials and Manufacturing Directorate, Air Force Research Laboratory (AFRL/RXCC), Wright-Patterson AFB, Dayton, OH 45433, USA

**Keywords:** carbon fibers, metal-organic frameworks, carbon nanotubes, additive manufacturing, mechanical characterization

## Abstract

Additively manufactured (AM) composites based on short carbon fibers possess strength and stiffness far less than their continuous fiber counterparts due to the fiber’s small aspect ratio and inadequate interfaces with the epoxy matrix. This investigation presents a route for preparing hybrid reinforcements for AM that comprise short carbon fibers and nickel-based metal-organic frameworks (Ni-MOFs). The porous MOFs furnish the fibers with tremendous surface area. Additionally, the MOFs growth process is non-destructive to the fibers and easily scalable. This investigation also demonstrates the viability of using Ni-based MOFs as a catalyst for growing multi-walled carbon nanotubes (MWCNTs) on carbon fibers. The changes to the fiber were examined via electron microscopy, X-ray scattering techniques, and Fourier-transform infrared spectroscopy (FTIR). The thermal stabilities were probed by thermogravimetric analysis (TGA). Tensile and dynamic mechanical analysis (DMA) tests were utilized to explore the effect of MOFs on the mechanical properties of 3D-printed composites. Composites with MOFs exhibited improvements in stiffness and strength by 30.2% and 19.0%, respectively. The MOFs enhanced the damping parameter by 700%.

## 1. Introduction

Short fiber reinforced polymer composites (SFRPs) comprise fibers with lengths ranging from microns to a few millimeters. Composites based on short fibers have numerous applications, including automotive interior structures, sports goods, and electrical circuit boards [1]. While SFRPs possess lower strength and stiffness than their continuous fiber counterparts, their mechanical performance is more tailorable through several parameters, including the fibers’ type, orientation, volume fraction, surface morphology, and interfacial adhesion to the matrix [2].

For most commercial applications, SFRPs are processed with thermoplastic matrices via extrusion or injection molding [1] and with thermoset matrices using injection-compression molding [3]. Typically, these methods are employed for geometrically simple structures such as sheets and tubes but are constrained by complex geometries and the need for molds. Additive manufacturing (AM) eliminates these constraints and facilitates a near-final-shaped production without a mold. Various methods of AM for SFRPs have been demonstrated, including fused deposition modeling (FDM) [4,5,6]. FDM carries several drawbacks; it uses a lamination-like technique to combine the laminae of a fused thermoplastic material. This induces delamination between the layers leading to shape distortion. The FDM minimum layer thickness is reported to be about 0.1 mm [4,5], allowing for the entrapment of significant air pockets between the deposited filaments. Alternatively, chopped fiber composites can be combined with epoxy matrices through a direct write method (DWM) [7,8]. This technique is essentially a concentrated viscoelastic fluid- (i.e., ink) deposition technique that relies on the ink strength for self-support of the printed structure. Compton and Lewis [7] utilized DWM for co-printing carbon fibers and SiC whiskers with an epoxy matrix. The printed composites significantly increased the modulus and strength over the neat resin. Identical results were reported by Pierson et al. [9], where the SFRP attained 66% of the theoretical tensile strength of aligned short fiber composite. The initial fibers’ average length was 6.00 mm, and after undergoing the ink processing technique, the length was reduced to 0.20 mm. The shortened fibers’ length is responsible for not achieving the expected theoretical strength value. Kanarska et al. [10] simulated the inkjet deposition process as a non-Newtonian fluid flow with a high fiber volume fraction. They found that when the fiber’s length exceeds two-thirds of the nozzle diameter, most fibers align around the nozzle walls, leaving a few randomly oriented fibers at the core of the deposited filament. They also concluded that the best alignment could be realized when the fiber’s average length is within 40–50% of the nozzle diameter. Classical micromechanical analysis of short fiber composites [11] suggests that the fibers must exceed a critical length proportional to the fiber’s strength and diameter and inversely proportional to the fiber/matrix bond shear strength. For carbon and glass fibers, this length is 20–150 times the diameter [1]. The critical fiber length restricts the AM of composites since most printers’ nozzles do not accommodate long fibers to avoid nozzle blockage. This compromise to ensure ease of manufacturing will reduce the strength of the composite. While the fiber strength and diameter are fixed, their bonding shear strength to the matrix can be modified by improving the fiber/matrix adhesion. Therefore, enhancing the adhesion allows for printing short fibers while yielding adequate strength of the composite. Fibers can be acid-treated or decorated with nanomaterials to improve adhesion to the matrix. Fu et al. [11] reported a noticeable improvement in the interlaminar shear strength after treating carbon fibers with acid. Using chemical vapor deposition (CVD), Yao et al. [12] grew CNTs on carbon fiber and observed good improvement in the interfacial shear strength. Metal-organic frameworks (MOFs) provide a feasible route to increase the surface area of carbon fibers. An MOF is a 3D structure encompassing metal ions coordinated with organic ligands [13]. These structures are naturally very porous with extreme surface areas [13]. The MOFs’ growth does not require a vacuum and is often carried out via wet chemistry at ambient conditions [14]. Currently, MOFs are used together with carbon materials for non-structural applications. Tran et al. [15] developed a porous catalyst combining CNTs and MOFs to detect urea. Yan et al. [16] coated graphene with MOFs forming a water-splitting electro-catalyst.

Incorporating MOFs into additive manufacturing was carried out by several research groups. For example, Kim et al. [17] developed an MOF photocurable fluoropolymer ink that can be sprayed, molded, or deposited via a pen to construct a water-tolerant sensor that can test for water pollutants. Lim et al. [18] utilized a direct ink writing technique to deposit a gel containing Cu_3_(BTC)_2_ nanoparticles to create a densely packed, self-standing MOF monolith. Another research group utilized a 3D printer to prepare several MOF-based solids with controlled morphologies from shear-thinning inks. All solids were structurally stable under 0.6 MPa of uniaxial compression [19]. Kearns et al. [20] offered a comprehensive review of shaping MOFs via different 3D printing techniques, including fused filament fabrication (FFF), digital light processing (DLP), selective laser sintering (SLS), and direct ink writing (DIW).

In this current investigation, we employed MOFs to alter the surface of short PAN-carbon fibers to enhance the interface shear strength. Additionally, we utilized the nickel-based MOF film as a catalyst to grow CNTs on the surface of short carbon fibers. Different mechanical and microstructural characterization techniques are implemented to reveal the performance and microstructure of these surface treatments.

## 2. Materials and Methods

The materials utilized in this investigation include EPON 826 (Miller Stephenson), CAB-O-SIL TS-720 fumed silica (Cabot Corp.), AS/BR102 chopped fiber (Hexcel, Inc., Decatur, AL, USA), and ethyl-methylimidazolium dicyanamide (ionic liquid, Sigma Aldrich, St. Louis, MO, USA). The fibers were de-sized inside a quartz tube furnace at 550 °C for 30 min under a nitrogen environment. The fibers were washed with acetone and DI water repeatedly and then dried at 100 °C in a convection oven for 24 hrs. To allow the formation of active sites (-COOH and NH_2_), the fibers were treated in a diluted mix (2:1) of de-ionized (DI) water: HNO_3_ acid for 24 h, rinsed thoroughly with DI water until a neutral pH of 7.0 was achieved, and finally dried at 100 °C for 24 h. To prepare the MOF growth solution, nickel nitrate hexahydrates (Ni (NO_3_)_2_·6H_2_O, Sigma Aldrich Co.) was dissolved in 100 mL of methanol to yield a 0.33 M concentration. The ligands solution dissolved 2-methylimidazole (C_4_H_6_N_2_, Sigma Aldrich Co., St. Louis, MO, USA) in 100 mL of methanol at 0.14 M concentration. Each solution was magnetically stirred separately at 400 rpm for 6 h. After this, the two solutions were mixed and magnetically stirred for 10 min. The chopped carbon fibers (5.0 mm long) were placed inside a mesh-open bag to keep them suspended with minimum flotation and immersed in the solution for 24 h. The carbon fiber bag was removed from the growth solution bath and washed repeatedly with ethanol. Then the fibers were removed from the bag and left to dry in an oven at 100 °C for 24 h. The CNTs growth was performed inside a quartz tube reactor outfitted with a thermal controller and three-input gas (N_2_, C_2_H_2_, and H_2_) mass flow controllers (MKS Co., Andover, MA, USA). The process begins with a reduction step to remove excessive oxides from the MOFs by flowing H_2_/N_2_ gas mixture atmosphere at 550 °C for two hours. Then, the tube reactor is flushed with N_2_ gas to eliminate residuals from the previous step. Subsequently, the CNTs growth step begins, maintaining the temperature of 550 °C for 30 min under a balanced C_2_H_4_/H_2_/N_2_ environment. The different carbon fibers were characterized using scanning electron microscopy (SEM, FEI Quanta 650, Thermo Fisher Scientific Inc Suwanee, GA, USA) equipped with energy-dispersive X-ray spectroscopy (EDS) detector (Bruker, Inc., Minneapolis, MN, USA), Transmission electron microscopy (TEM, FEI Talos, Thermo Fisher Scientific Co.) at an acceleration voltage of 200 kV was utilized to examine the CNTs morphology and size and their growth topology. Wide-angle X-ray scattering (WAXS) was performed using Xeuss 3.0 (Xenocs Inc, Holyoke, MA, USA). The X-ray wavelength was 1.54 Å, the sample-to-detector distance was 55 mm, and the exposure time was two h. A Pilatus3 300 k detector was used to collect scattered X-rays. Two-dimensional scattering patterns were analyzed using the Datasqueeze software [21]. All 2D scattering patterns were integrated over a 90° wedge parallel to the fiber axis to create the 1D scattering charts. Scattering normal to the fiber axis was excluded from the integration. To reveal the functional groups resulting from the different fiber surface treatments, Fourier transform infrared spectroscopy (FTIR, Nicolet 380) was carried out at wavelengths of 600–4000 cm^−1^ by employing the attenuated total reflection (ATR) mode. The thermal integrity of the reference and different treated fibers was investigated using thermogravimetric analysis (TGA, TA Q500). from RT to 900 °C at 10 °C/min under an air environment.

Printable inks were prepared for each of the different fiber treatments. For each configuration, approximately 20 g of EPON 826 was poured into a 150 mL cup. After each addition, 10 parts per hundred (pph) of fumed silica were added in 2 equal increments and mixed with the epoxy in a Thinky ARV-310 planetary mixer under vacuum for 3 min at 2000 rpm after each addition. Five pph of a fiber configuration was added gradually to the slurry and mixed for three sets of 3 min at 2000 rpm. Then, 5 pph of the ionic liquid was added, followed by another round of mixing at 2000 rpm for 1 min. Identical procedures were repeated for the four fibers configuration: reference, acid treated, coated with MOFs, and sheathed with CNTs.

The composite samples were printed using a nScrypt 3Dn-500 printer (nScrypt, Inc., Orlando, FL, USA). The prepared inks were loaded into 10 cm^3^ Nordson syringes and centrifuged at 3000 rpm for 10 min to reduce entrapped air, then transferred to a fresh syringe using a high-pressure adapter to push the ink from the filled syringe into the empty one. A 0.58 mm diameter nozzle was attached to the syringe. Print speed was calibrated by measuring the ink mass flow rate at pressures of 44–52 psi. The pressure setting on the Nordson Ultimus V pressure pump was 11–13 psi, and the high-pressure adapter supplied a pressure amplification factor of 4. The mass flow rate was divided by the nozzle diameter and ink density to calculate a linear extrusion rate which was used as the gantry translation speed or print speed. The road-to-road spacing between the deposited ink filaments was equal to the nozzle diameter. An optical profilometer was used to measure the height of one layer, which was set as the next layer’s height onward. After printing, samples were cured at 100 °C for 15 h using a convection oven. Two sets of samples were printed for each fiber configuration, a tensile sample with a dog-bone shape following the ASTM638, sample type-V.

The tensile tests were carried out using MTS Criterion™ Model 43 system (MTS, Inc., Eden Prairie, MN, USA) equipped with a 5.0 kN load cell and a digital image correlation system (DIC, Correlated Solutions, Inc., Irmo, SC, USA) to measure the strain. The sample tests were extended at a 1.0 mm/min speed until failure was achieved. A total of 6 samples were tested for each configuration. The dynamic mechanical analyzer (DMA8000, Perkin Elmer, Inc., Waltham, MA, USA.) was utilized following the ASTM D7028 and ASTM D5023 standards. Composite coupons of 50.0 mm × 4.0 mm× 1.0 mm were used. A 40 mm span 3-point bending fixture was utilized to mount the samples. During the temperature sweep, a constant frequency of 1.0 Hz and a constant force of 1.0 N were utilized while ramping the temperature from 25–200 °C. Isothermal frequency sweep tests were held at 25 °C while increasing the frequency from 1–100 Hz under a constant force of 1.0 N.

## 3. Results and Discussion

The micrographs in Figure 1 show the evolution of the carbon fiber surface through the different treatments. The de-sizing of the fibers under a nitrogen environment peeled off the sizing films, as shown in Figure 1a. The etching of the fiber in the nitric acid, Figure 1b, removed the sizing residue. After MOF growth, Figure 1c, the fiber surface was fully sheathed with porous cells separated by nano walls morphologies. These morphologies tremendously increased the fibers’ surface area compared to the reference or acid treated. This rough morphology of the nickel-based MOF is unique and different from the smooth films obtained via magnetron-sputtering pure nickel metal on carbon fiber [22]. Consequently, the CNT growth, as shown in Figure 1d, was very dense as there are far more surfaces for the catalytic reaction of Ni with the hydrocarbon gas to grow CNTs. The TEM micrographs, inset Figure 1d, revealed that the CNTs are multi-walled with an average diameter of 15–20 nm, and have non-uniform multi-walls, suggesting these MWCNTs incorporate several defects. The growth at a relatively lower temperature is a possible cause for such wall defects, as high temperatures could anneal them [23].

Energy dispersive X-Ray spectroscopy (EDS) was utilized to inspect the Ni and C elements in MOFs and the CNTs in the composites. Figure 2a shows the elemental micrographs of a representative spectrum of the areas for a sample representing composites with MOFs, while Figure 2b depicts the EDS analysis upon reducing the fiber coated with MOFs under an inert environment at 550 °C. Figure 2c reveals the EDS elemental analysis of a fiber after growing CNTs on its surface. Table 1 summarizes the statistics of the EDS analysis of different spots on the different fibers samples. All three samples exhibit peaks for crystalline Ni; however, the sample after the reduction steps shows the strongest Ni peak indicating that most of the organic components of the MOFS have decomposed upon the reduction of the fiber.

The WAXS patterns in Figure 3 revealed a diffuse anisotropic scattering. The neat reference carbon fiber and the acid-treated fiber exhibited a fan-like scattering observed for AS4 fibers by other researchers [24]. This pattern is imputed to the random distribution of defects on the fiber’s surface The similarity of the scattering patterns for the functionalized and reference fibers implies that the acid treatment did not significantly alter the fiber’s crystallinity. 

It was established that PAN-based carbon fibers develop several functional groups, such as -COOH and -NH_2_, on their surfaces when immersed long enough in nitric acid [25]. These groups are vital for anchoring the metal seeds to the fiber’s surface and the consequent MOF growth.

The FTIR transmittance spectra of the various fibers’ configurations are presented in Figure 4. The reference AS4 carbon fibers’ FTIR spectra do not exhibit significant peaks for the absence of functional groups. It is also inferred that the ATR technique may not observe the minuscule changes attributed to de-sizing the fibers at elevated temperatures in the I.R. spectrum due to inadequate resolution [26]. The nitric acid oxidation of the desized fibers resulted in the emergence of several peaks. The groups of C-O-C, C-O-N, or C-N appear between 1400 cm^−1^ and 1000 cm^−1^ [27]. These peaks suggest that the nitric acid introduced oxygen functional groups to the surface of the fibers. The broad peak between 3200–3600 cm^−1^ is associated with the OH. group stretching vibrations [28], indicating water presence due to insufficient drying of the fibers. Several strong peaks, not observed in the other fibers treatments, emerged upon the deposition of MOFs on the fibers. The strong peaks beyond 3400 cm^−1^ indicate the OH group coupling to Ni(ii). The peaks around 1600 and 1200 cm^−1^ are correlated to the stretching of coordinated carboxylate (-COO) and symmetric stretching mode of coordinated carboxylate, respectively [29]. The peaks from 750–800 cm^−1^ are associated with the O-Ni-O vibrations, while those around 1000 and 800 cm^−1^ substantiate the C-N and C-H bonds, respectively [30].

The TGA analysis results in Figure 5 show that the different samples started degrading at different temperatures. The reference fibers showed the slowest degradation rate starting at 600 °C. This behavior is consistent with other research groups for the T650 fibers [31]. The acid-treated fibers showed the next slowest degradation rate denoting that the surface groups yielding from acid oxidation did not add much weight to the fibers. The fibers with surface grown CNTs start degrading as early as 450 °C and exhibit 2 inflection points at 475 °C and 575 °C. The early degradation is mainly attributed to the oxidation of CNTs and disintegration of the amorphous carbon and defective CNTs. Indeed, it is a standard procedure to utilize moderate temperature to purify CNTs by oxidation [32]. One can also observe that the disintegration of the CNTs took place within the temperature range for their growth, identical to the findings by other research groups [33]. Two inflection points imply that different carbon species degrade at different temperatures; the earlier inflection point is associated with the disintegration of CNTs, while the second inflection point signifies the disintegration of the fibers. The carbon fibers for this sample degrade faster as they have been exposed to elevated temperatures during the reduction step and during the growth of the CNTs, which explains their fast degradation. The fibers sheathed with MOFs attained the second fastest degradation rate starting at 550 °C. The inadequate drying can explain the early weight loss of this sample since the sample was immersed in an acidic solution for 24 h, followed by soaking in a methanol bath for 24 h during the MOF growth stage. Thus, this sample accumulated a significant amount of moisture, which was not eliminated by oven drying. Furthermore, the disintegration of the organic ligaments of the MOF can contribute to weight loss.

Figure 6a displays sample tensile tests of composites based on the different fiber treatments, while the average strengths and moduli of these composites are shown in Figure 6b. All composites demonstrated a linear elastic behavior. All different surface treatments reduced the ductility of the composites manifested by the shortened strain-to-failure spans. However, all the various surface treatments improved the stiffness and strength compared to the reference composite. The acid treatment improved the strength and stiffness by 7.7% and 16.3%, respectively. The acid treatment induces nanoscale roughness on the surface of the fiber. Previous investigations [34,35] illuminated the effect of HNO_3_ treatment of PAN carbon fibers attributing similar improvements to increasing the surface free energy by more than 40% due to establishing C-N polar groups, improving the chemisorption of the resin and curing hardener on the carboxylic groups at fibers surface, complemented by an increase in the surface area. Oxidizing carbon fibers by diluted acid treatment can remove surface irregularities or misaligned sheaths on the fiber’s outer surface and lead to better bonding to the epoxy matrix due to the presence of functional groups [36]. The individual contribution to each of these effects is an ongoing research topic.

Growing the MOFs on the fibers led to the most significant improvement in strength and modulus by 19.0% and 30.2%, respectively, above the reference desized composite values. The Ni-MOF topology amplifies the surface area bestowing a novel mechanical anchoring mechanism between the fibers and the matrix. One rationale is that the pits and pores on the surface of the fibers, caused by the acid, were occupied with MOF. Unlike pure metal, the dual effect of the -COOH group induced by the acid etching and the organic ligand in the MOFs facilitates good anchoring of the Ni-MOF structure to the activated sites on the fiber surface. The anchored MOF promote enhanced load transfer from the matrix to the fiber, increasing the enduring load capacity and the interfacial shear strength yielding a better tensile strength. The noticeable increase in the modulus arises from sheathing the fiber with an outer layer of stiff Ni-MOF structures, acting like a different material than that based on the bare fibers. Growing MOFs on carbon fiber is still relatively new; the closest effort involves electrolytic Ni-plating of carbon fibers [37]. The Ni-plating was observed to successfully enhance the surface polarity by establishing oxygenated functional groups.

Exploiting the Ni in the MOFs as catalyst led to the successful growth of CNTs. This growth introduced a minute improvement in strength by 3.0%. This marginal improvement originates from several factors. First, the growth process of CNTs entails high temperatures that could remove the surface groups gained by acid treatment and MOF deposition. This temperature also destroys the external graphitic layers of the carbon fiber. Second, the same growth temperature, considering that the growth did not require a vacuum, damages the fibers. Uniform CNT coating on the carbon fibers often yields a modest strength improvement [38]. Finally, unlike the MOF, the CNTs bond to the fiber surface is less intense, evidenced by the observation that most CNTs disperse in the matrix when they undergo the intense mixing processes for preparing the printer ink. We believe that the existence of a residue of MOF layer beneath the CNTs film still contributes to increasing the strength. This layer of MOF residue could explain the modulus increase by 14.0%.

The morphologies of the fracture surfaces of the various composites were examined using SEM, as displayed in Figure 7. The composite made with de-sized fiber revealed significant gaps between the epoxy and the fibers, as can be seen in the inset of Figure 6a which signifies a weak adhesion between the two constituents. This debonding was lessened per the acid treatment of the fibers because of the existence of new surface groups, as shown in Figure 6b. Upon growing MOFs on the fibers, the fibers did not display noticeable deboning, as seen in Figure 6c. This is primarily due to the strong bonds between the MOF legend component and the epoxy and fibers.

The mechanical interlocking due to the encapsulation of the epoxy matrix in the pores of MOF cells also contributes to the strengthening of the adhesion of the matrix to the fiber. Growing CNTs had the same effect on eliminating the debonding, and a portion of the matrix is seen attached to the surfaces of the pulled-out fiber (Figure 6d) indicating a strong adhesion between the fiber, CNTs, and the matrix.

Figure 8 depicts the results of the DMA analysis of the various composites. The figures display the loss tangent parameter, tan (δ), which portrays the ratio of the composite’s ability to dissipate energy over its ability to store energy. The glass-transition temperature (Tg) was deduced from the peak of the tan (δ) curves. It is evident that the Tg values did not alter much by the different surface treatments of the fibers; changes were within 4 °C at the most. We expected that the combination of carbon fibers and CNTs could yield a noticeable shift in Tg by hindering the polymer chain mobility, as we had noticed for composites with 60% wt. plain woven continuous carbon fibers [39]. However, it is inferred from the current study that MOFs and CNTs do not affect the mobility of epoxy chains in the glass transition region. This could be ascribed to the low carbon fiber weight percentage. Identical phenomena were reported by other research groups [30,31], which were ascribed to epoxy polymerization and the diminishing of the functional groups when the temperature approaches the Tg [40,41].

The DMA frequency sweep results are displayed in Figure 8b, manifested by the variation in the damping factor, tan (δ). In general, the damping capacity of composites can be elucidated by a collection of different mechanisms, including stick-slip friction, interfacial debonding, and nanofillers [42]. The reference fibers composite exhibited the smallest damping parameter. Functionalizing the fibers with nitric acid improved tan (δ) by 20% at 90 Hz. One hypothesis to explain this improvement is that the acid etching enhanced the surface area enough to facilitate a larger contact area with the epoxy, facilitating more significant energy dissipation.

Furthermore, the acid-damaged graphitic fiber surfaces could have also stimulated the stick-slip mechanism at the interface. Growing MOF on the fiber surface increased the surface area tremendously. Furthermore, the delicate metal/epoxy bonding permits additional energy dissipation; thus, tan (δ) increased by 700% at 85 Hz over the reference fiber composite. The CNTs increase the surface area significantly due to the extreme aspect ratios of the CNTs. This tremendous interfacial area facilitates significant friction between the CNT/epoxy interfaces generating considerable energy dissipation [22]. The composite with CNTs enhanced the damping coefficient by up to 400% at 85 Hz.

## 4. Conclusions

This work outlines a feasible yet scalable path for modifying the interfacial morphologies between short carbon fibers and polymeric matrices. Several mechanical properties of composites based on short carbon fibers (including strength, stiffness, and damping parameter) were significantly enhanced by growing MOF on the fibers prior to composites fabrication via the DWM additive manufacturing technique. The MOF amplify the interfacial area between the epoxy and the fibers and facilitate better adhesion. FTIR analysis verified that the chemical groups induced by the MOF synthesis contribute to the fiber’s surface functionalizing. The synthesis of MOF also substantiates catalytic Ni seeds suitable for CNT growth. The growth of MOF and CNT is anticipated to add multiple functionalities to the composite, including enhanced thermal and electrical transport properties. Tensile test results indicate that MOF’s strength enhancement surpasses any other treatment, including CNTs. This improvement results from the significant surface area furnished by the porous MOF coupled with mechanical anchoring of the epoxy within the MOF pores. The contact angle analysis confirmed that fibers with nanoscale features, such as MOs and CNTs, exhibit better wettability. Furthermore, the FTIR analysis has shown that the MOF treatment of the fibers generated several chemical groups that have not been discerned in other surface treatments. The hybrid fiber/MOF composite also achieved better viscoelastic damping performance than all other composites, including those based on CNT surface treatment, yielding a tan delta improvement of 700%.

## Figures and Tables

**Figure 1 nanomaterials-13-00944-f001:**
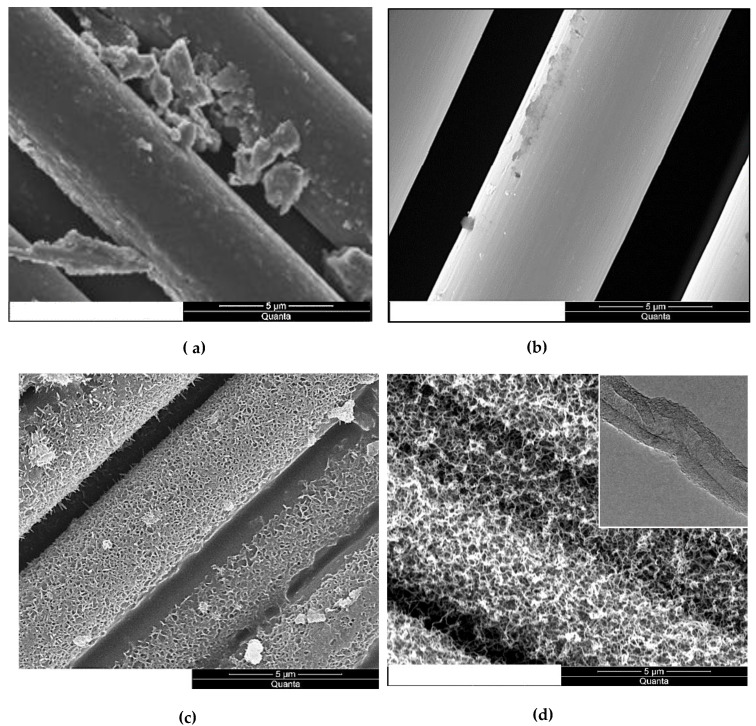
SEM micrographs of (**a**) AS4 de-sized fibers, (**b**) acid oxidized fibers, (**c**) fibers with MOFs growth, and (**d**) fibers with CNTs growth utilizing MOFs as a catalyst. (Inset) TEM micrograph of isolated CNT grown utilizing Ni-MOFs as a catalyst.

**Figure 2 nanomaterials-13-00944-f002:**
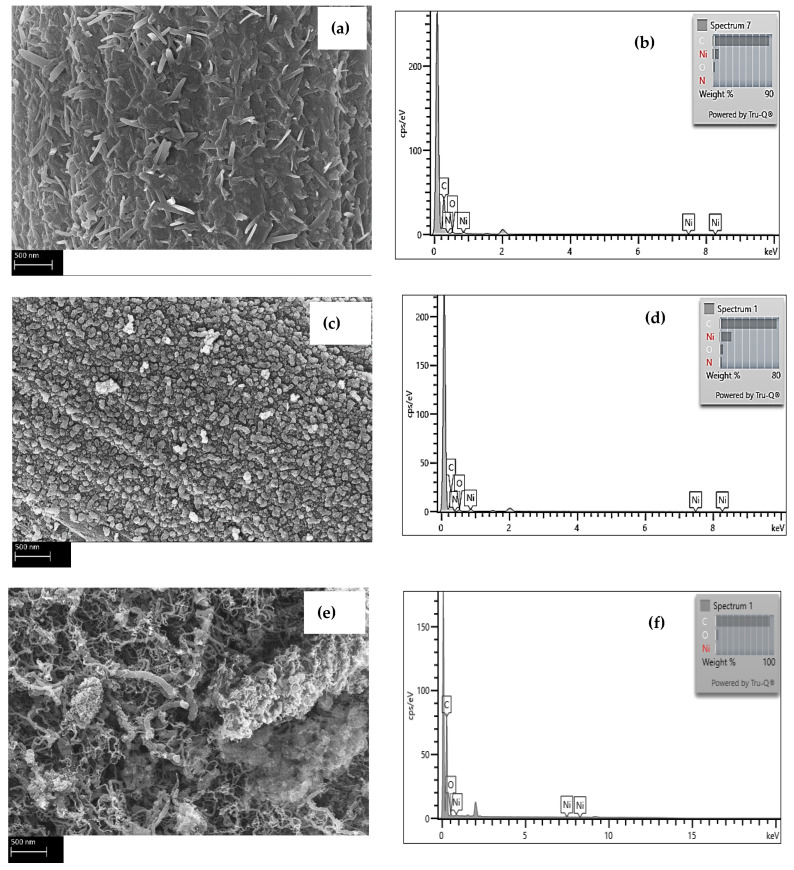
EDS of C and Ni elemental analysis for (**a**,**b**) carbon fiber with MOFs, (**c**,**d**) fibers with MOFs after reduction in a furnace under an inert environment, and (**e**,**f**) carbon fiber/CNTs reinforcement. The scale bar is 1 micron.

**Figure 3 nanomaterials-13-00944-f003:**
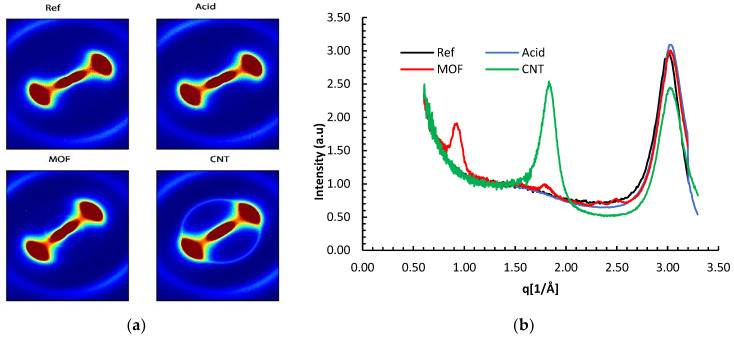
(**a**) Two-dimensional WAXS patterns of the different carbon fibers configurations, and (**b**) WAXS profiles of the AS4 carbon fibers after the different surface treatments.

**Figure 4 nanomaterials-13-00944-f004:**
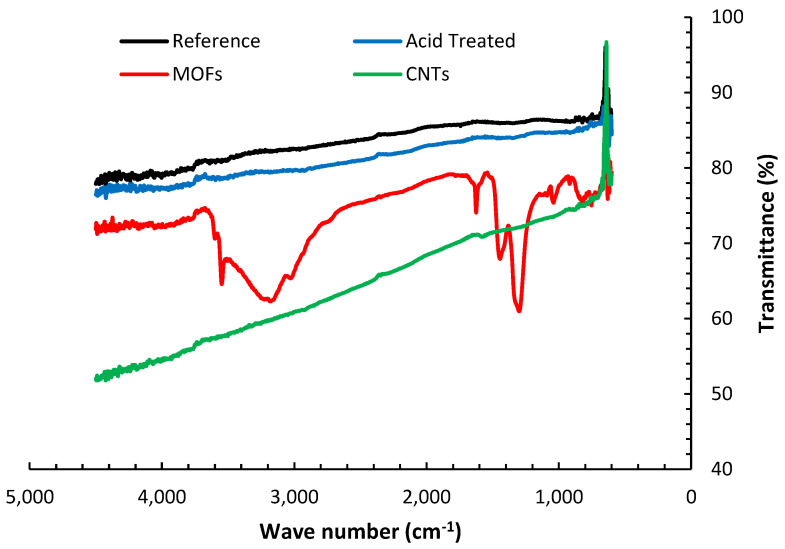
FTIR spectra of carbon fibers after different surface treatments.

**Figure 5 nanomaterials-13-00944-f005:**
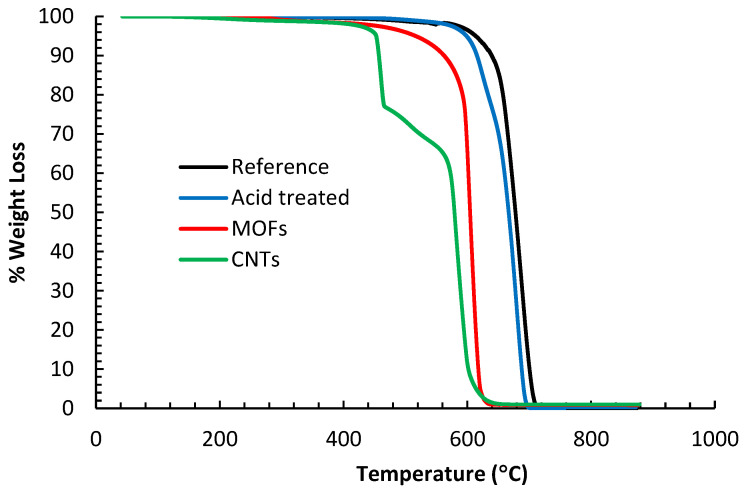
TGA profiles the reference desized AS4 carbon fibers, fibers after acid oxidation, fibers with MOF growth, and fibers with surface-grown CNTs.

**Figure 6 nanomaterials-13-00944-f006:**
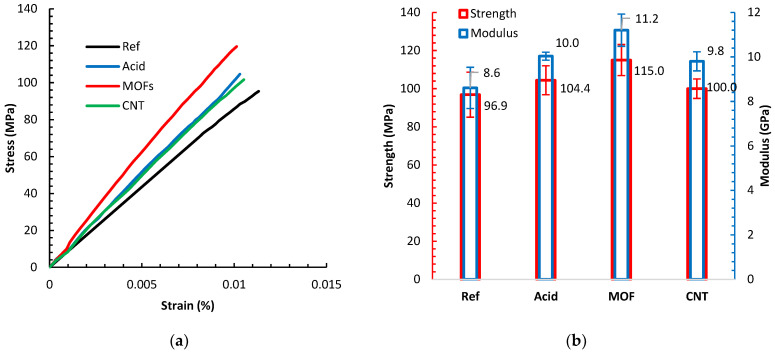
(**a**) Sample tensile tests for the composites with various fiber treatments. (**b**) The averaged stiffness and tensile strength of the different composites. Error bars denote the standard deviation.

**Figure 7 nanomaterials-13-00944-f007:**
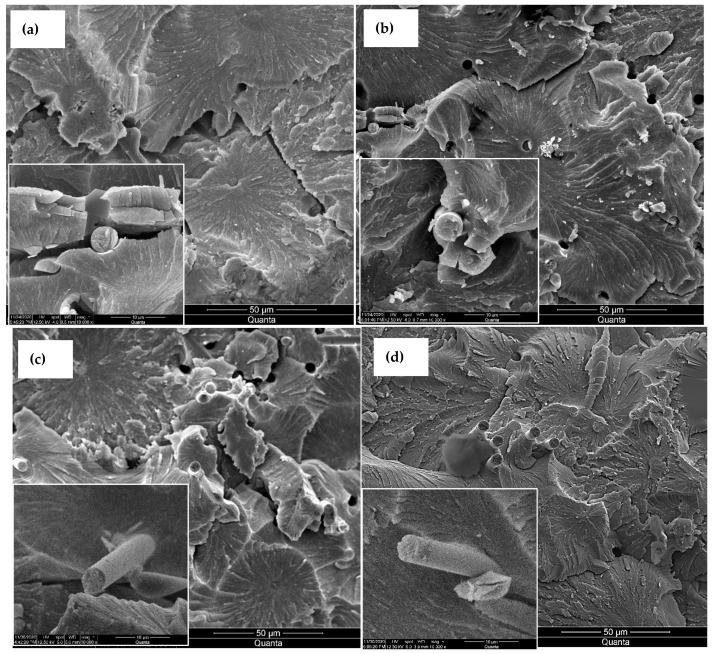
SEM fractography at different magnifications for composites based on (**a**) reference desized fibers, (**b**) acid-oxidized fibers, (**c**) fibers with surface-grown MOFs, and (**d**) fibers with CNTs. Inset images scalebar 10 μm.

**Figure 8 nanomaterials-13-00944-f008:**
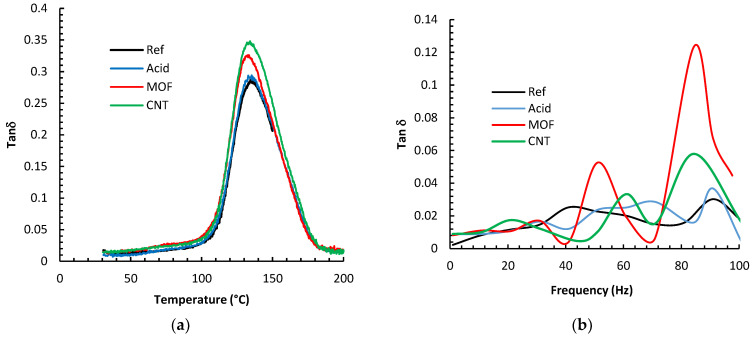
(**a**) DMA measurements of the damping parameter; tan (δ) under temperature sweep for the various composites based on different fibers treatments. Measurements were conducted at a 1.0 Hz constant frequency. (**b**) The DMA measurements of the damping parameter for the various composites via frequency ramping at room temperature.

**Table 1 nanomaterials-13-00944-t001:** EDS elemental analysis of carbon fibers with MOF, carbon fibers with MOF after reduction under an inert environment, and carbon fibers after growing CNTs.

	MOFs			Reduced MOFs			CNTs		
Element	C	O	Ni	C	O	Ni	C	O	Ni
Average	90.65	3.81	4.54	73.08	4.35	15.54	96.19	1.78	2.03
Standard Deviation	1.11	1.05	0.67	4.54	0.38	2.14	0.50	0.41	0.66

## Data Availability

Data is contained within the article.

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
