# Peer review of "Hybrid Metal-Organic Frameworks/Carbon Fibers Reinforcements for Additively Manufactured Composites"

_nanomaterials, 2023, doi:10.3390/nano13050944_

Round 1

Reviewer 1 Report

This work demonstrated the design and fabrication of additively manufactured composites composed of a MOF and the grown CNT on the fiber. The idea is very interesting and novel. The work is in general well organized. But the characterizations of materials are relatively weak. Major revisions are needed to address the following comments before the publication of this work in Nanomaterials.

1.      A paragraph should be added into the Introduction to introduce the use of additive manufacturing to solidify the highly porous MOF materials. The following important literatures from different research groups should be cited in this paragraph: ACS Appl. Mater. Interfaces 2019, 11, 4, 4385–4392; ACS Mater. Lett 2019, 1, 147–153; Adv. Funct. Mater. 2019, 29, 1805372; ACS Appl. Mater. Interfaces 2020, 12, 9, 10983–10992; J. Mater. Chem. A 2021, 9, 27252-27270; ACS Appl. Mater. Interfaces 2022, 14, 24, 28247–28257.

2.      XRD patterns of all the fiber before growth, MOF-grown fiber, and CNT-grown fiber must be reported in the manuscript to verify the crystallinity and phase purity of the MOF. Simulated powder XRD pattern of the same Ni-based MOF downloaded from the crystallographic database should be plotted together as well to compare. This result must be added in the manuscript.

3.      Elemental analysis (either EDS, ICP or XPS is fine) should be performed to verify the presence of the Ni-based MOF, and the result should be included in the manuscript.

4.      This one is optional, but if possible, the authors should perform nitrogen adsorption-desorption experiments to measure the isotherms of these fibrous materials. The porosity of each material can thus be probed and reported in the manuscript. This characteristic is also very important for MOF-based materials.

5.      The authors claimed that MOF is the “catalyst” for CNT growth. However, a temperature of 550 degree C was used to remove metal oxide clusters from the MOF and used for MOF growth. This temperature should be obviously higher than the degradation temperature of the MOF. Then there should not be any MOF anymore after the CNT growth. The XRD data in comment#2 should confirm this point. If this is the case, can the authors define MOF as the “catalyst”? A “catalyst” needs to return to its initial status after the reaction and should be recyclable. I think the authors may need to rephrase it.

Reviewer 2 Report

Dear authors,

The proposed work is interesting and I propose it for publication. It has been well organized and the structure is well. I ask you to implement the following highlight to improve the quality of the prepared work:

1. Abstract: It is proposed to talk more quantitatively about the obtained results in order to address the applied characterization techniques (DMA, TGA, etc.).

2. Introduction: When talking about Additive Manufacturing, I propose to talk in 2-3 lines about the different techniques and/or different applicable materials. Please also include the following references as well:

https://doi.org/10.3390/polym14173674   https://doi.org/10.3390/ma15248722

https://doi.org/10.3390/ma15207193

3. Materials and methods: The applicable materials and utilized methodology have been well explained and all the required information are included.

4. Results and discussion: it is section 3 (not 2) and please kindly revise it.

5. Figure 1 and 6: The scale bar of the SEM images are not clear and it should be revised.

6. Figure 2-a: is it possible to add an explanation or a caption to the figure? There is no information on the images.

7. Figure 3: In the FTIR curve, it is possible put the vertical axe on the other side and it would be more clear.

8. Figure 6: In the SEM images, it would be better if you include some remarks on the images such as the existence of fibers, mode of deformation, etc.

Best wishes,

Round 2

Reviewer 1 Report

Comments have been addressed thus the paper can be published.

Reviewer 2 Report

The paper could be published

Best wishes,